

# Explaining the green volcanic sunsets after the 1883 eruption of Krakatoa

Christian von Savigny[1], Anna Lange[1], Christoph G. Hoffmann[1], and Alexei Rozanov[2]

[1]Institute of Physics, University of Greifswald, Felix-Hausdorff-Str. 6, 17489 Greifswald, Germany
[2]Institute of Environmental Physics, University of Bremen, Otto-Hahn-Allee 1, 28359 Bremen, Germany

**Correspondence:** Christian von Savigny (csavigny@physik.uni-greifswald.de)

**Abstract.** Volcanic sunsets are usually associated with extended and enhanced reddish colours typically complemented by purple colours at higher elevations. However, many eyewitnesses reported remarkably clear and distinct green twilight colours after the eruption of Krakatoa (Sunda Strait, Indonesia) on August 27, 1883. To our best knowledge, no earlier studies exist providing an explanation for this unusual phenomenon. In the current work we employ simulations with the SCIATRAN radiative transfer model to investigate the processes leading to green volcanic sunsets. Green sunsets can be simulated based on plausible assumptions by anomalous scattering on stratospheric sulfate aerosols. We investigate the sensitivity of the twilight colours to relevant parameters such as aerosol optical depth, the parameters of the particle size distribution and the amount of ozone. The main requirements for the occurrence of green twilights are a sufficiently large aerosol optical depth combined with particle radii of about $500 - 700\,\mathrm{nm}$ (assuming stratospheric sulfate aerosols) and a preferably narrow aerosol particle size distribution. The occurrence of green twilights after historic eruptions provides important constraints on the particle size of volcanic aerosols.

## 1 Introduction

Volcanic eruptions are well known to affect the colours of the twilight sky. In conditions of volcanically enhanced stratospheric aerosol loading the primary sunset colouring is complemented by a secondary effect, also known as "afterglow", lasting significantly longer than ordinary twilight colours (Meinel and Meinel, 1991). The extended reddish afterglows reported after many volcanic eruptions can be qualitatively explained in single scattering approximation. The tangential path of the solar radiation through the lower atmosphere (the sun being below the horizon) leads to a reddening of the transmitted solar spectrum due to Rayleigh scattering and extinction by aerosol particles. This red spectrum is then scattered by the volcanically enhanced aerosol layer to produce a typically reddish afterglow for a ground-based visual observer. The red colouring of the evening sky typically intensifies with increasing stratospheric aerosol loading. In a few studies, this behaviour was exploited to estimate aerosol optical depth from red-to-green colour ratios in photographs of historic paintings (Zerefos et al., 2007, 2014). This approach was recently challenged (von Savigny et al., 2022) and one important aspect was the dependence of volcanic twilight colours on the size distribution of the volcanic aerosol particles.



The Royal Society's report on the Krakatoa eruption in 1883 (Symons et al., 1888) includes numerous reports on green
twilight skies after the eruption. For example, the Hon. F. A. Rollo Russell observed on November 9, 1883 (from Surrey, i.e.
south of London) a "remarkable whitish-greenish opalescence above the sun at sunset [..]". On November 25 he observed
"green light above sunset, and bright greenish-white growing from about 10 minutes after sunset". Others reported "very
curious opalescent shining green and slightly greenish-white" or "bright green glow near the place where the sun set". The
Krakatoa report also includes observations of the typical pink or violet colours often associated with volcanic sunsets, but the
reports of green colours are remarkable. Unusual colours (blue and green) of the solar and lunar disks are sometimes reported
after biomass burning events, sandstorms or volcanic eruptions (see, e.g. Horvath et al. (1994) or Wullenweber et al. (2021)),
including the eruption of Krakatoa (Symons et al., 1888). This phenomenon is relatively well understood and can be reproduced
using radiative transfer models (Wullenweber et al., 2021). The physical cause is most likely anomalous scattering, i.e. the
scattering cross section increases with increasing wavelength in the visible spectral range. Anomalous scattering requires a
specific aerosol particle size distribution with particle radii in the range of $500 - 700$ nm (assuming the refractive index typical
for stratospheric sulfate aerosols) and a preferably narrow distribution. Anomalous scattering also provides an explanation for
the green twilight skies in the aftermath of the Krakatoa eruption, as shown in the current study. A fundamental difference
between blue/green appearances of the solar or lunar disk and green twilights is that in the first case the transmitted radiation
is seen by the ground-based observer, whereas in the second case it is scattered radiation.
This paper is structured as follows. The employed radiative transfer simulations and colour display approach are explained
in section 2. Section 3 presents the main findings of our study. In section 4 important implications of the results are discussed
and conclusions are presented in the final section.

## 2   Methods

The methodology employed here is very similar to that used in the recent studies by von Savigny et al. (2022) and Lange et
al. (2023) and only the most important details are repeated here. Spectra of scattered solar radiation are simulated with version
4.1.3 of the radiative transfer model SCIATRAN (Rozanov et al., 2014) for a ground-based observer viewing the twilight sky.
The calculations are carried out for solar zenith angles (SZAs) between $90°$ and $98°$, variable viewing zenith angles (VZAs,
see Table 1), and a fixed solar azimuth angle (SAA) of $0°$. See Fig. 1 for an illustration of the viewing geometry. SCIATRAN
was operated in the approximate spherical mode, i.e. single scattering calculations are carried out in fully spherical geometry,
while the contribution of multiple scattering is approximated. This approximation, however, is accurate enough to investigate
twilight conditions for the range of SZAs considered here (Lange et al., 2022). Table 1 provides an overview of the range of
parameters chosen for the SCIATRAN simulations.
The volcanic stratospheric aerosol layer considered here has a Gaussian particle number density profile with a peak altitude
of $z_{peak} = 21$ km and a FWHM (full width at half maximum) of 8 km. The peak altitude was chosen based on estimates of the
layer height reported in Symons et al. (1888) (see Part IV, Section IV, Table IV) for the period between one and three months
after the eruption of Krakatao. Note that the number density profile is scaled within SCIATRAN to result in the specified AOD.





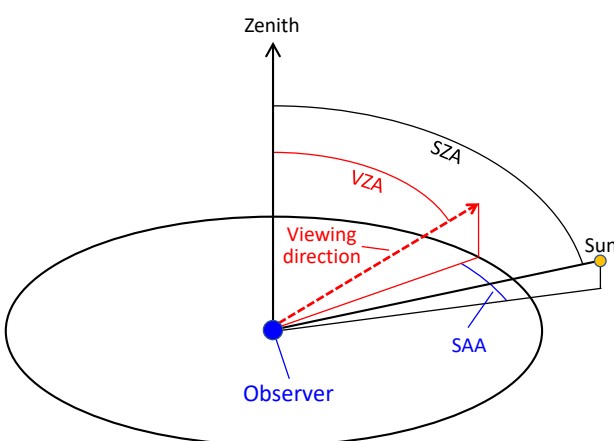

**Figure 1.** Sketch of the observation geometry and the relevant angles (i.e. solar zenith angle, SZA; viewing zenith angle, VZA; solar azimuth angle, SAA).

The stratospheric aerosol is assumed to consist of pure sulfate particles composed of 75 % $H_2SO_4$ and 25 % $H_2O$. The aerosol particle size distribution (PSD) is assumed to be mono-modal and log-normal with median radius $r_{med}$ and geometric width parameter $S$, both of which will be varied in the following. The assumption of a pure sulfate layer in the stratosphere can be considered reasonable several months after a major volcanic eruption (when many of the green twilight observations were made after the Krakatoa eruption (Symons et al., 1888)), because the volcanic ash can be assumed to have sedimented out (Kremser et al., 2016). Note that the tropospheric aerosol component is also assumed to consist of sulfate aerosol particles with the same PSD as the stratospheric component. The tropospheric AOD corresponds to about 10 % of the total AOD. Tests were also performed based on the tropospheric aerosol model by Shettle and Fenn (1979), which had a minor overall effect on the simulation results.

In order to ensure an objective display of the colours corresponding to a given radiance spectrum, we use the CIE (International Commission on Illumination) colour matching functions and determine the x and y chromaticity values (see, e.g. Wullenweber et al. (2021) for more detailed information on the display of colour information). Colour information is then displayed in the x-y plane (see, e.g. the right panel of Fig. 2). The symbols displayed in the CIE chromaticity diagrams in Figs. 2, 3, and 4 are filled with the RGB colours corresponding to the respective values of x and y. Please note that only the position within the x-y plane constitutes objective colour information, the displayed colour does not necessarily, because it will depend, e.g. on the output device.

## 3 Results

We first present a sample scattering twilight spectrum and the corresponding CIE chromaticity diagram for a parameter combination that is associated with distinctly green twilight colours. The upper row of Fig. 2 shows the scattered radiance spectrum



**Table 1.** Overview of parameter settings for the SCIATRAN simulations ($^\dagger$ Note that SAA $= 0°$ corresponds to the solar direction)

| Parameter | Abbreviation/Variable | Values |
|---|---|---|
| Solar zenith angle | SZA | $90° - 98°$ in steps of $1°$ |
| Viewing zenith angle | VZA | $0° - 88°$ ($5°$ steps for $[0°, 70°]$; $2°$ steps for $[70°, 88°]$) |
| Solar azimuth angle$^\dagger$ | SAA | $0°$ |
| Aerosol optical depth | AOD | $0 - 0.3$ in steps of $0.05$ |
| Median radius | $r_{med}$ | $150$ nm $- 850$ nm in steps of $100$ nm |
| Distribution width | S | $1.05, 1.1, 1.2, 1.4, 1.5, 1.6$ |
| Total ozone column | TOC | $200 - 400$ DU in steps of $50$ DU |

(left panel) and the corresponding CIE chromaticity diagram (right panel) for the following parameters: SZA $= 93°$, VZA $= 88°$, SAA $= 0°$, AOD $= 0.3$, $r_m = 650$ nm, S $= 1.1$, and TOC $= 400$ DU. The spectrum exhibits a maximum in the green spectral range (which covers wavelengths from about 495 nm to about 570 nm) and the Chappuis absorption bands of $O_3$ (with a maximum near 600 nm) are clearly visible.

The right panel shows the location of the spectrum in the CIE chromaticity diagram as a coloured circle. The arc of smaller coloured circles shows the locations of pure spectral colours, while the red "x" corresponds to the unattenuated solar spectrum. The resulting colour for this combination of viewing geometry and input parameters is clearly greenish, demonstrating that green twilight colours can be simulated with suitable chosen input parameters. This is already an important finding of the present study. The lower row of Fig. 2 shows results for the same viewing geometry as in the upper row, but without any

aerosols in the atmosphere, i.e. AOD $= 0.0$. In this case, there is no clear spectral maximum in the green part of the spectrum, and the orange/red parts are significantly enhanced relative to smaller wavelengths in the visible spectral range (Note the different ordinate ranges in the two spectra).

    The visual colour impression of the evening sky depends on several different parameters, including the viewing and solar angles, the parameters of the aerosol PSD and the abundance of ozone. We found that green sunsets can be simulated for many

different parameter combinations and a complete description of all possible combinations is beyond the scope of the present study. Instead, the main effects are investigated here in a sensitivity analysis by varying the following parameters: SZA, VZA, the PSD parameters $r_{med}$ and $S$, as well as AOD and TOC. Figure 3 shows CIE chromaticity diagrams for evening sky spectra for SZA $= 91°$ and variable VZA for different values of $r_{med}$ (panel a), TOC (panel b), AOD (panel c) and S (panel d). The default values of the parameters that are not varied in a given panel are: AOD $= 0.3$, $r_{med} = 650$ nm, S $= 1.1$, and TOC $=$

300 DU. These values were chosen, because they lead to green twilights for certain viewing geometries. For reasons of clarity the chromaticity coordinates are only shown for two TOC values (panel b) and two values of the distribution width S (panel d).

    As panel (a) of Fig. 3 shows, the median radius of the aerosol PSD has a relatively large impact on the resulting twilight colours and only for $r_{med} = 650$ nm, greenish colours occur for VZAs $\geq 80°$, with VZAs $= 88°$ being associated with a rather yellowish colour. For smaller values of $r_{med}$ anomalous scattering is not strong enough – or does not occur at all – and the

twilight colours are rather orange and reddish. The impact of the TOC on the twilight colours is already quite significant at



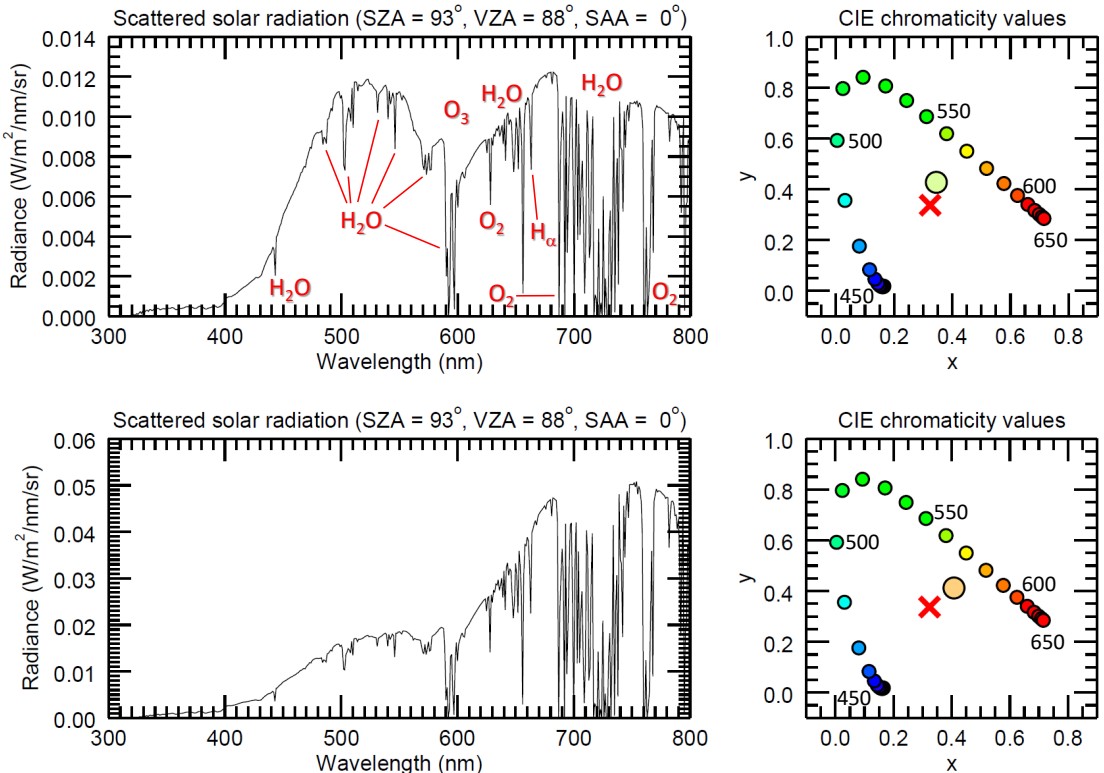

**Figure 2.** Spectra (left panels; note different ordinate ranges) and CIE chromaticity diagrams (right panels) for the following viewing geometry: SZA = 93°, VZA = 88°, SAA = 0°. The simulations shown in the top row are based on the following assumptions: AOD = 0.3, $r_{med}$ = 650 nm, S = 1.1, TOC = 400 DU. For the bottom row, the AOD has been set to zero and the other parameters remain the same. The "x" in the right column panels corresponds to the chromaticity values of the unattenuated solar spectrum. The arc of colour circles corresponds to pure spectral colours at different wavelengths (values given in nm with 10 nm steps) and the single coloured circle shows the chromaticity values of the spectrum on the left.



SZA = 91°, as panel (b) of Fig. 3 shows. Panel (c) demonstrates that large AODs provide more favorable conditions for green twilights to occur. Increasing the AOD moves the chromaticity values to the left in the CIE chromaticity diagram. Panel (d) displays the dependence of the twilight colours on the width parameters $S$ of the aerosol PSD. Apparently, green sunsets occur more easily for narrow PSDs. This is also according to expectations, because anomalous scattering is more pronounced for
narrow PSDs – given the right values of $r_{med}$ (compare Fig. 1 in Wullenweber et al. (2021)).

The resulting colours also depend on the SZA and Fig. 4 shows results similar to Fig. 3 for SZA = 93°. Overall, the dependence of the colours on the parameters varied in the respective panels is generally larger for SZA = 93° than for SZA = 91°. Based on all the simulations carried out, the following conclusions can be drawn:

–    Green twilight skies can only be simulated for specific aerosol sizes (see panels (a) in Figs. 2 and 3). The median radius
of the assumed log-normal PSD needs to be between about 500 nm and 700 nm. This size range is expected, because it is
        associated with the most pronounced anomalous scattering for stratospheric sulfate aerosols (Wullenweber et al., 2021).

–    Absorption in the Chappuis bands of $O_3$ is only of moderate importance for SZAs smaller than about 94°. For larger
        SZA the effect of the $O_3$ absorption on the twilight colours increases (not shown). For increasing TOC, the absorption
        in the Chappuis bands will be more pronounced, causing the spectral maximum in the blue/green part of the visible
spectrum to shift to lower wavelengths.

–    Green twilights can only be simulated for AOD values exceeding about 0.2 for the parameters chosen. The fact that
        a sufficiently large AOD (in combination with a suitable PSD) is required to produce green twilight colours can be
        explained qualitatively, because the red/orange colours caused in a pure Rayleigh (or Rayleigh-dominated) atmosphere
        need to be suppressed.

–    Green twilight skies are in general more easily produced for narrow PSDs, as seen in panel (d) of Figs. 3 and 4. This
        behavior is also expected, because anomalous scattering is more pronounced for narrow PSDs (e.g. Fig. 1 in Wullenweber
        et al. (2021)). An increase in the width of the PSD leads mainly to a horizontal shift of the chromaticity coordinates to
        the right, i.e. in the direction of red and orange colours.

It is also relevant to note that the impact of the surface albedo on the twilight colours is very small (see Fig. 4 in von Savigny
et al. (2022)).

## 4    Discussion

It is important to realize that not all possible combinations of relevant input parameters affecting the colours of the twilight sky can be tested. For that reason we cannot give, e.g. a sharp AOD threshold above which green twilights will occur. In addition, the exact aerosol PSD and its variation after the 1883 eruption of Krakatoa is unknown. Based on OPC (Optical Particle Counter)
measurements after the 1991 eruption of Mt. Pinatubo (Deshler, 2008) one might expect the PSD after the Krakatoa eruption to be bi-modal with one mode at a median radius of about 100 nm and a coarse mode with median radii of several hundred nm.





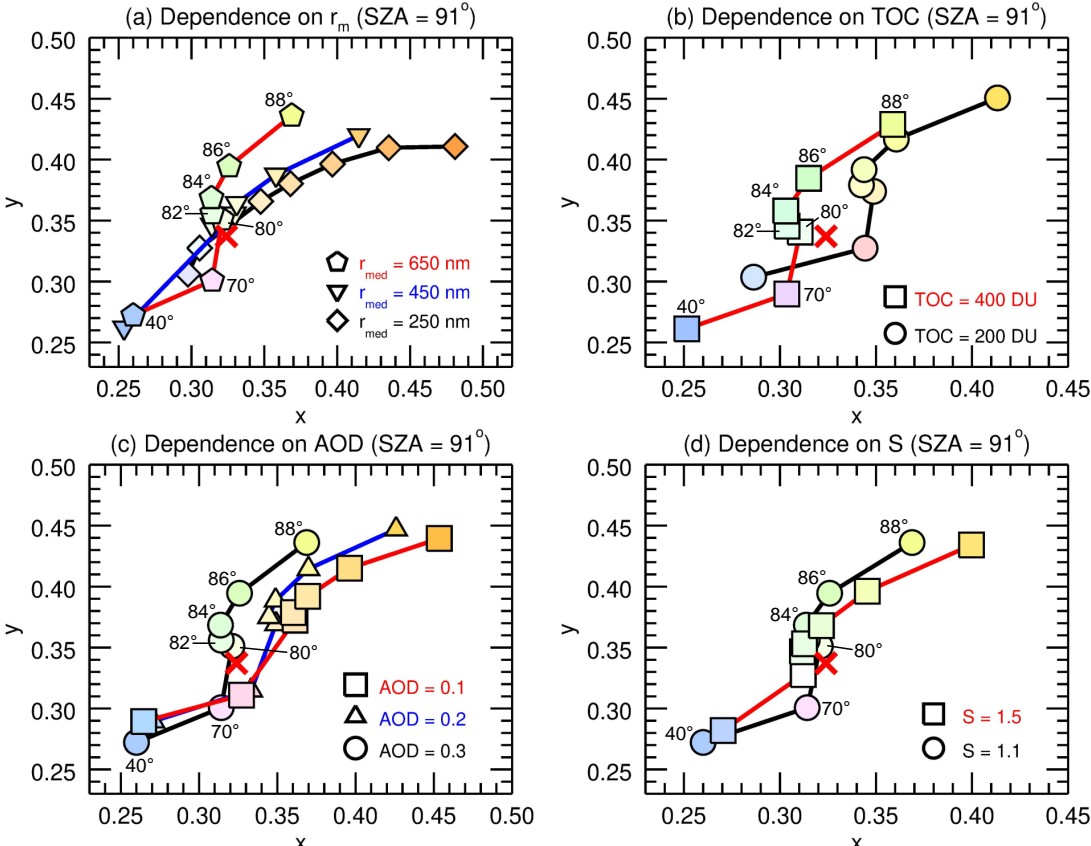

**Figure 3.** CIE chromaticity diagrams similar to the right panel of Fig. 2 but for variable input parameters, i.e median radius $r_{med}$ (panel a), AOD (panel b), total ozone column TOC (panel c), and width $S$ of the PSD. Results are for SZA = 91°, SAA = 0°, and different VZAs, shown by the numbers next to the symbols in each panel (VZA values are 40°, 70°, 80°, 82°, 84°, 86°, and 88°). For reasons of clarity, VZAs are only shown for one line in each panel.

However, because of the strong dependence of the particle's scattering cross section on radius (von Savigny and Hoffmann, 2020), the scattering signal can be expected to be strongly dominated by the coarse mode, providing some justification for the assumption of a mono-modal PSD. In addition, the main purpose of the present study is to demonstrate that green volcanic

sunsets can in principle be simulated with suitably chosen parameters that are in a plausible range, without knowledge of the actual aerosol PSD and its spatial and temporal evolution after the eruption of Krakatoa.

In the following we discuss – based on the information available for Krakatoa or other eruptions – whether the AOD and particle size parameters in the aftermath of the Krakatoa eruption were likely in the range required to explain green volcanic sunsets. Regarding the particle size of stratospheric aerosols, typical radii of stratospheric sulfate particles are 100 nm (Deshler,

2008; Wrana et al., 2021) for background conditions. After the Mt. Pinatubo eruption in June 1991 particle radii of several hundred nanometers were reported, with some studies reporting radii on the order of 600 nm up to several months after the





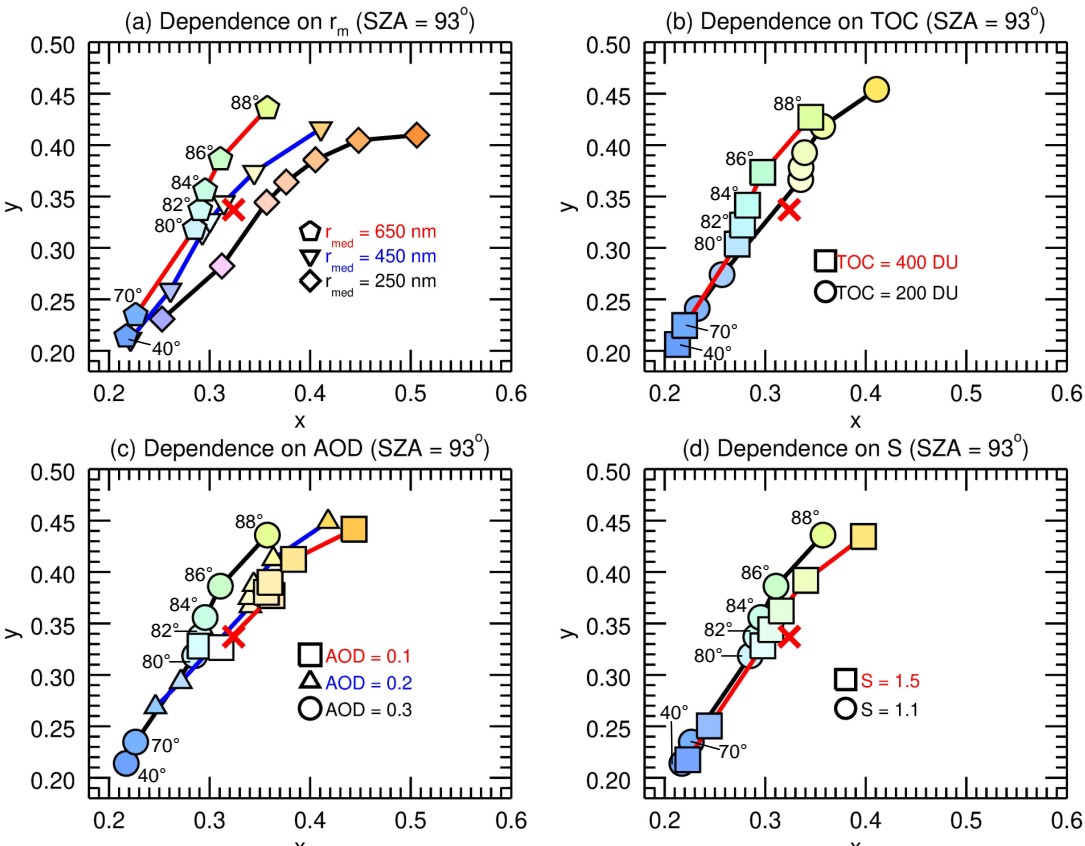

**Figure 4.** Similar to Fig. 2 but for SZA = 93°.

eruption (Brock et al., 1993; Echle et al., 1998; Bingen et al., 2004; Deshler, 2008). In other words, particle radii on the order of the ones required to produce green volcanic sunsets have also been observed after the Mt. Pinatubo eruption. We are not aware of any reports of green sunsets after this eruption, which may be due to (a) the AOD being too low or (b) the PSD being

too wide. Evidence for the possibility of particles with radii in the 500 nm to 700 nm range also comes from model simulations. Kokkola et al. (2009) tested the performance of different aerosol microphysical modules and found – for a Pinatubo-sized eruption – a coarse mode of the PSD with a diameter between 1 $\mu$m and 2 $\mu$m, i.e. radii in the range required to cause green volcanic twilights.

Next we discuss available estimates of the AOD following the 1883 eruption of Krakatoa, which are based on sunlight

extinction measurements (Sato et al., 1993; Stothers, 1996). According to Sato et al. (1993), the AOD reached a maximum globally and annually averaged value of 0.125 only in 1885. The values in 1883 and 1884 are 0.021 and 0.081, respectively. A maximum AOD in 1885 appears implausible and Sato et al. (1993) make no statements about the accuracy of the AOD values. Stothers (1996) reports a maximum AOD of 0.4 in September 1883 for the [30°S, 30°N] latitude range. For January 1884 a globally averaged AOD of 0.182 is reported. However, the discussion in Stothers (1996) suggests that these values



are relatively uncertain. Since no precise measurements of the AOD are available for the period 1883 – 1885, we can only carefully conclude that the AOD may have been in the range for which our simulations produce green twilight skies. The global mean SAOD (stratospheric AOD) values reconstructed from sulfate records in ice cores by Toohey and Sigl (2017) reach a maximum of about 0.12 after the Krakatoa eruption, while the ICI (ice core index) by Crowley and Unterman (2013) yields a maximum global mean SAOD of about 0.18. It is of course important to remember that for certain times and latitude regions the SAOD will exceed the global mean value. Summarizing, the AOD and particle size required for green volcanic sunsets are in a plausible range for the Krakatoa eruption and more or less consistent with earlier studies. Our results suggest that the AOD in the aftermath of the 1883 eruption of Krakatoa may have been larger than the values presented in some of the earlier studies.

It is also interesting to note that for scenarios that produce a green sunset, the usual orange and red colours of the twilight sky can be suppressed or even completely absent (see, e.g. panels (a) of Figs. 3 and 4). This is consistent with some of the eyewitness reports cited in Symons et al. (1888). One report states (referring to the year 1884): "During the remainder of the year the sunsets were uncommonly free from colour, [...], and no redness of an ordinary character remaining along the horizon after sunset, except on a few evenings and in a few localities." (Symons et al., 1888).

Next we discuss the relationship between green volcanic sunsets and the recently described phenomenon of the green band (Lange et al., 2023), corresponding to a typically narrow horizontal band of light green colour sometimes occurring after sunset between the red colours near the horizon and the blue colours above (Lange et al., 2023). Green volcanic sunsets are characterized by much more pronounced green colouring of the twilight sky with significantly larger distances from the white point in the CIE chromaticity diagram (compare, e.g. Fig. 4 in this study with Fig. 4 in Lange et al. (2023)). In addition, the green band does neither require the large particles necessary for anomalous scattering to occur nor the relatively large AODs (Lange et al., 2023). Also, in case of the green volcanic sunsets simulated here, the green colour extends to the horizon, in contrast to the green band phenomenon.

## 5 Conclusions

In the aftermath of the 1883 eruption of Krakatoa numerous eyewitnesses reported green twilights. In this work, we carried out sensitivity studies based on radiative transfer simulations in order to investigate the cause of green twilights and the sensitivity of twilight colours to different relevant parameters. Green volcanic twilights can be explained by anomalous scattering occurring for sufficiently large particles (i.e. radii of about $500\,\mathrm{nm}$ to $700\,\mathrm{nm}$) and a preferably narrow particle size distribution. In addition, green twilights require the AOD to exceed a threshold of about 0.2. The dependence of twilight colours on AOD, the particle size parameters and the abundance of $O_3$ was generally found to increase with SZA for SZAs between $90°$ and $98°$. Due to the fact that different parameters affect the twilight colours, we cannot give precise values of the AOD or the size parameters for the occurrence of a green volcanic twilight. Despite these limitations, green twilights can provide some quantitative constraints on the AOD and aerosol size for historic volcanic eruptions and complement other volcanic proxy data sets, e.g. based on ice cores or lunar eclipse observations.



*Code and data availability.* The SCIATRAN radiative transfer model including all data bases used for the simulations presented in this work is publicly accessible at https://www.iup.uni-bremen.de/sciatran/ (last access: July 20, 2023).

*Author contributions.* CvS designed the study, carried out the SCIATRAN simulations with assistance by AR and AL and wrote an initial version of the manuscript. All authors discussed, edited and proofread the manuscript.

*Competing interests.* The authors declare that they have no competing interests.

*Acknowledgements.* This work was supported by the Deutsche Forschungsgemeinschaft (project VolARC of the DFG research unit VolImpact FOR 2820, grant no. 398006378). We are indebted to the Institute of Environmental Physics of the University of Bremen – particularly to
195 Dr. Vladimir Rozanov and Prof. Dr. John P. Burrows FRS – for providing the SCIATRAN radiative transfer model.



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
