# Peer review of "Explaining the green sunsets after the 1883 volcanic eruption of Krakatoa"

_EGUsphere, 2023_

## Author Comment (AC1)

**Reply to comments by anonymous reviewer #2**

**Comment:** The paper describes a small study which explains why there can be green sunsets after volcanic eruptions. Radiative transfer model simulations are performed for various different aerosol assumptions to explain the green color and to also constrain stratospheric aerosol parameters, i.e. size distribution and aerosol optical thickness. The detailed discussion shows that the found parameters are reasonable. The authors suggest that the method could be used to constrain aerosol properties of historic eruptions, however I suppose that this would be difficult because there are not many reports about the sky colors available and of course also no quantitative observations.

The paper is clearly presented and since it explains the phenomenon of green sunsets to my knowledge for the first time I recommend to publish the paper after very minor corrections (see below).

*Reply: We thank the reviewer for his/her positive assessment of our manuscript!*

**Comment:** Often, the term "anomalous scattering" is used. I think that this is misleading because the dependece of the scattering coefficient on wavelength is just a result of well-known Mie theory and not "anomalous".

*Reply: We agree that "anomalous scattering" may be misleading, but this is a standard technical term for the case, where the scattering cross section increases with increasing wavelength. It has been used in the past (e.g. Porch et al., 1989) and was not introduced by us. For this reason we would like to keep the term. However, in order consider the reviewer's comment and to clarify this point, we added one sentence to the introduction stating that Mie theory can also be used to simulate anomalous scattering.*

*Reference:*

*Porch, W. M., Blue moons and large fires, Appl. Opt., 28, 10, 1989.*

**Comment:** It is not mentioned how the optical properties of the aerosols were computed. I assume Mie theory?

*Reply: The reviewer is right, this was not mentioned. Instead, we did refer to earlier papers based on a similar approach. But it is certainly an important piece of information and is now mentioned in section 2.*

**Comment:** The term "volcanic sunset" in the title sounds strange. To me "Explaining the green sunsets after the 1883 volcanic eruption of Krakatoa" sounds better.

*Reply: OK, changed.*

---

## Author Comment (AC2)

**Reply to review by Filip Vanhellemont**

The authors provide an explanation for the observed green twilight observations that were reported by eye witnesses after the eruption climax of Krakatoa in 1883. They use the well-known SCIATRAN radiative transfer code to simulate radiances, based on an educated assumption on the aerosol density profile shape. The perceived color is then estimated by converting radiances to chromaticity values using the CIE color matching functions. The color is studied as function of particle size distribution parameters, total ozone column and aerosol optical depth, for a number of solar zenith angles representative of twilight. The authors arrive at the clear conclusion that green twilight can be simulated for sufficiently large aerosols from a narrow size distribution and sufficiently large optical depth.

General Comments

This article is perfectly suited for publication in ACP. I am not aware of any other publication that presents an explanation of volcanically related green twilight, so the obtained conclusions are important. Furthermore, while it is difficult to obtain precise numbers on e.g. particle size from observations that are subjective (visual color perception), clear constraints/thresholds are obtained on particle size and optical depth. These are new numerical results on an important past volcanic event, and should be published.

> *Reply: We thank the reviewer for his positive evaluation of our manuscript!*

The applied methodology is clearly explained, and it should be possible for other researchers to reproduce the results. The article title and the abstract perfectly describe the content. The body text is very well written and needs almost no adaptation. Plenty of references are provided.

I really don't see any major problem with this paper; I have a few minor remarks that should be taken into account (see below), so I strongly recommend the publication of this paper.

> *Reply: Thank you!*

Specific comments

**Comment:** It is not clear to me at which wavelength Aerosol Optical Depth (AOD) is evaluated in the entire paper. This should be specified.

> *Reply: Thanks, the wavelength should of course be mentioned. It is now mentioned in Table 1.*

**Comment:** The finding that green twilight is associated with narrow size distributions suggests that these green colors are only observed in the early stages of the stratospheric aerosol evolution (say, a few months after the event of August 27). Coagulation of particles shifts the distribution to higher size values, but also tends to widen the distribution, but this process takes quite some time. It might explain why no green twilight is reported at later times after the eruption (as far as I can tell from the paper). Perhaps you can add a small comment if you agree with this.

> *Reply: Thanks for this suggestion. We are, however, not sure how the particle size distribution (PSD) will exactly evolve after an eruption. In another one of our recent papers (Wrana et al., 2023; reference provided below) we investigated the evolution of the PSD after the eruptions of Ambae (2018), Ulawun and Raikoke (2019) and La Soufriere (2021). In three*

*of theses cases, the median (and effective) radius of the stratospheric aerosols – as retrieved from SAGE III/ISS solar occultation measurements – decreased rather than increased as a consequence of the eruption. In addition, the reduction in size seen in the observations lasted for about half a year after the eruption of Ulawun. The accompanying model simulation also reproduced the initial decrease in size (effective radius), but after August 2019 the modelled sizes increased again, probably due to coagulation. Also because of the discrepancy between observation and model we are uncertain what role coagulation actually plays and how effective and fast it is to change the PSD significantly. For the Raikoke eruption (2019) an increase in particle size is seen in both observations and model simulations. In summary, we do not really know what exactly could have happened after the Krakatoa 1883 eruption and decided not to mention the hypothesis proposed by the reviewer. But it is certainly a possible interpretation or explanation.*

*Reference:*

*Wrana, F., Niemeier, U., Thomason, L. W., Wallis, S., and von Savigny, C.: Stratospheric aerosol size reduction after volcanic eruptions, Atmos. Chem. Phys., 23, 9725–9743, doi.org/10.5194/acp-23-9725-2023, 2023.*

**Comment:** In Fig. 3, the phenomenon of purple light that is often observed after large volcanic eruptions is also simulated, but it is not mentioned in the paper. A small description (one sentence or so) of this finding would enhance the credibility of the method even more.

*Reply: Another good point, thanks for pointing this out. The purple colours are now mentioned in section 3.*

**Technical Corrections**

**Comment:** Line 3: volcanic activity of Krakatau started already several months before August 27. It is perhaps better to speak about the eruption climax of August 27, or something similar.

*Reply: Good point. We decided to remove the exact date from the abstract, because otherwise the sentence would suggest that the green sunsets were observed immediately after the eruption. Now only the year of the eruption in mentioned.*

**Comment:** Line 70: This sentence seems to be incorrect. Suggestion: 'Please note that …, while the displayed …'.

*Reply: Changed.*

**Comment:** Figure 3: the caption seems to be wrong. Panel (b) shows Total Ozone Column, Panel (c) Aerosol Optical Depth, while the caption indicates the reverse.

*Reply: Thanks for catching this – corrected.*

---

## Author Comment (AC3)

**Reply to comment by Timothy Garrett**

There is a similar phenomenon of green sunsets that is seen for the case of storm clouds as discussed by Bohren and Fraser (1993) https://journals.ametsoc.org/view/journals/bams/74/11/1520-0477_1993_074_2185_gt_2_0_co_2.xml. Specifically, Bohren argues that the color of such sunsets owes to a blue-shift of red sunset light due to a combination of scattering and a preferential absorption by condensed water in the red. Can this explanation be excluded for the case described here? A substantial amount of condensed water would be required, but of course Krakatoa was an exceptional event.

*Reply: This is an interesting suggestion, thank you! We considered the hypothesis that the green colours could be caused by liquid water or ice in clouds carefully and concluded that it cannot provide an explanation for the green sunsets after the Eruption of Krakatoa for the following reasons:*

1. *The total tropospheric H2O mass is about 13 Tt (Tera tonnes) or 1.3 x 10^7 Mt. The mass of H2O injected by Krakatoa into the atmosphere is not known, but it may be estimated based on available estimates of the emitted mass of S and typical relationships between H2O and S in volcanic emissions. Some studies suggest that about 15 Tg of S were emitted by Krakatao in 1883, which corresponds to about 30 Tg of SO2. If we conservatively assume a H2O/SO2 mass ratio of 50, the volcanic H2O emissions would correspond to 1500 Tg or 1500 Mt of H2O, i.e. about 0.01 % of the troposphere's H2O mass. Tropospheric H2O is replaced about 40 times per year, i.e. it seems very unlikely that the H2O emitted by Krakatoa led to a significant increase in the occurrence of green thunderstorms several months after the eruption.*
2. *The hypotheses by Bohren and Fraser require really large cloud optical depths, as mentioned several times in Bohren and Fraser (1993). With such thick clouds twilight phenomena would not occur. This corresponds to an observation that can be frequently made: under overcast conditions, coloured sunsets do not occur.*
3. *The hypotheses by Bohren and Fraser provide explanations for green thunderstorms only for scenarios where the sun is above the horizon, while the remarkable reports of green colours mainly occur for twilight conditions, i.e. the sun is already below the horizon.*

*But it is interesting that your summary, i.e. "a blue-shift of red sunset light" still hits the mark. In case of the green volcanic sunsets, the solar radiation reaching the aerosol layer has been reddened due to Rayleigh extinction and is then "anomalously" scattered by aerosol particles of a suitable size distribution, i.e. scattering by aerosols leads to a blue shift in the spectrum, different from the hypothesis by Fraser and Bohren in Bohren and Fraser (1993).*

*We added a paragraph in the section 4 (discussion) to discuss the relationship of green sunsets and green thunderstorm clouds.*

---

## Author Response (AR2)

**Reply to editor comment by Dr. Tim Garrett**

Dear Dr. von Savigny,

Thank you for your response to the reviews on your intriguing paper. Please provide within the response file not only the reviewer comment and your response, but also the verbatim changes made in the manuscript, at least on the more significant points. This facilitates the review process and clarifies publicly the procedure within the open access review framework.

>*Reply: Sorry for this omission. We have now added the verbatim changes to the manuscript to the replies to the reviewer and editor comments (see below).*

Moreover, I would like to see a more considered response to the suggestion I made about considering a possible analogy to green thunderclouds. While the clouds of the Krakatoa explosion may not have been as rich in water as a thunderstorm over the Great Plains, the droplets were about 100 times smaller, and the value of (1-g) where g is the asymmetry parameter was about 50% higher as well. Based on the discussion in e.g. Bohren and Clothiaux starting p. 442, this would suggest that the clouds could have a water path that is about 10 times smaller than that of a thunderstorm, and still produce green sunsets. Such values may still be too thick but the current discussion in the revised paper does not make a convincing case that the explanation can be omitted. The comparison to Hunga Tonga - Hunga Ha'apai seems out of place given how much smaller it was by comparison in its total explosive power.

Best regards,

Tim Garrett

>*Reply: Thank you for the information and the suggestions that we considered carefully.*
>
>*It is correct that we have some particle size and AOD estimates, but it is important to note that they are for stratospheric aerosol particles and not for water clouds. Typical radii of stratospheric aerosols are on the order several hundred nm, while typical radii of cloud droplets are on the order of tens of microns. One might argue that the hypothesis by Bohren to explain green thunderstorms may also be applied to stratospheric aerosol particles to explain green sunsets. However, the available OD estimates (in part based on direct measurements) of the stratospheric aerosol layer after the 1883 Krakatoa eruption are all lower/equal than 0.4, i.e. much lower than the OD values of a thunderstorm cloud. We also estimated the "liquid water path" of an enhanced stratospheric aerosol layer and it is many orders of magnitude lower than the liquid water path required for a green thunderstorm cloud, according to Bohren. This is not surprising, because the total mass of stratospheric sulfate aerosols (even after a major volcanic eruption) is at least 6 orders of magnitude lower than the $H_2O$ mass in the troposphere.*
>
>*We also used equation 8.80 in the textbook by Bohren and Clothiaux (2006), inserted values of g (asymmetry parameter) and d (particle diameter) typical for stratospheric aerosols after a major volcanic eruption and got about a 75% reduction of the liquid water required to produce green colours (according to the hypothesis by Bohren). Given the low overall mass of the stratospheric aerosol layer, this will not allow explaining green colours caused by stratospheric aerosols, however.*

*So, the problem is that the OD and size estimates we have are valid for stratospheric aerosols (not clouds) and the stratospheric aerosol OD is very small compared to the OD of a thunderstorm cloud. We can essentially exclude the possibility that the Bohren hypothesis (applied to stratospheric aerosols) explained the green twilights after the Krakatoa eruption. However, they may of course have been occurrences of green thunderstorm clouds after the eruption of Krakatoa. Several of the green twilight observations described in the Krakatoa report explicitly mention that the sun was visible during sunset or before the sun has set, i.e. the twilight sky was cloud free and the green colours cannot be explained by green thunderstorm clouds.*

*We would like to mention again that the summary in your previous editor comment, i.e. "a blue-shift of red sunset light" still hits the mark and provides a qualitative explanation of all the known "green" sky colour effects. In case of green volcanic sunsets, the solar radiation reaching the aerosol layer has been reddened due to Rayleigh extinction and is then "anomalously" scattered by aerosol particles of a suitable size distribution, i.e. scattering by aerosols leads to a blue shift in the spectrum, different from the hypothesis by Fraser and Bohren in Bohren and Fraser (1993).*

*We added the following paragraph in the section 4 (discussion) to discuss the relationship of green sunsets and green thunderstorm clouds:*

*"Green sky colours are also sometimes observed in connection with severe thunderstorm clouds (e.g. Bohren and Fraser, 1993) and we briefly discuss similarities and differences between green thunderstorms and green sunsets. Bohren and Fraser (1993) discuss two hypotheses to explain green thunderstorm clouds, i.e. the "Fraser" hypothesis and the "Bohren" hypothesis. The Bohren hypothesis is essentially based on the absorption of solar radiation by liquid water (or ice) inside thunderstorm clouds, leading to a blue-shift of sunlight reddened by Rayleigh-extinction for large SZAs. Both hypotheses require very large cloud optical depths. One might argue that the Bohren hypothesis may also be applied to stratospheric aerosols, because the smaller particle size of stratospheric aerosols compared to cloud droplets is associated with a reduction in the liquid water path required to explain green thunderstorm clouds according to the hypothesis by Bohren (see section 8.6.1 and equation 8.80 in Bohren and Clothiaux (2006)). However, because of the much lower optical depth of the stratospheric aerosol layer after the Krakatoa eruption (see above) compared to typical thunderstorm clouds, the Bohren hypothesis cannot explain green twilights caused by the stratospheric aerosol layer after the Krakatoa eruption. It is certainly possible that there was an increased occurrence of green thunderstorm clouds after the eruption of Krakatoa, but in several of the green twilight observations described in the Krakatoa report the twilight sky was cloud free and green thunderstorms cannot serve as an explanation of the green colours.*

*However, a similar general process leads to the green colours in case of both thunderstorms and volcanic sunsets, i.e. a blue-shift of reddened sunlight. This blue-shift is caused by Rayleigh-scattering in case of the Fraser-hypothesis, by $H_2O$ (liquid or solid) absorption in case of the Bohren hypothesis and by anomalous aerosol scattering in case of the green volcanic sunsets."*

**Reply to review by Filip Vanhellemont**

The authors provide an explanation for the observed green twilight observations that were reported by eye witnesses after the eruption climax of Krakatoa in 1883. They use the well-known SCIATRAN radiative transfer code to simulate radiances, based on an educated assumption on the aerosol density profile shape. The perceived color is then estimated by converting radiances to chromaticity values using the CIE color matching functions. The color is studied as function of particle size distribution parameters, total ozone column and aerosol optical depth, for a number of solar zenith angles representative of twilight. The authors arrive at the clear conclusion that green twilight can be simulated for sufficiently large aerosols from a narrow size distribution and sufficiently large optical depth.

General Comments

This article is perfectly suited for publication in ACP. I am not aware of any other publication that presents an explanation of volcanically related green twilight, so the obtained conclusions are important. Furthermore, while it is difficult to obtain precise numbers on e.g. particle size from observations that are subjective (visual color perception), clear constraints/thresholds are obtained on particle size and optical depth. These are new numerical results on an important past volcanic event, and should be published.

> *Reply: We thank the reviewer for his positive evaluation of our manuscript!*

The applied methodology is clearly explained, and it should be possible for other researchers to reproduce the results. The article title and the abstract perfectly describe the content. The body text is very well written and needs almost no adaptation. Plenty of references are provided.

I really don't see any major problem with this paper; I have a few minor remarks that should be taken into account (see below), so I strongly recommend the publication of this paper.

> *Reply: Thank you!*

Specific comments

**Comment:** It is not clear to me at which wavelength Aerosol Optical Depth (AOD) is evaluated in the entire paper. This should be specified.

> *Reply: Thanks, the wavelength should of course be mentioned (it is 550 nm). It is now mentioned in the caption of Table 1.*

**Comment:** The finding that green twilight is associated with narrow size distributions suggests that these green colors are only observed in the early stages of the stratospheric aerosol evolution (say, a few months after the event of August 27). Coagulation of particles shifts the distribution to higher size values, but also tends to widen the distribution, but this process takes quite some time. It might explain why no green twilight is reported at later times after the eruption (as far as I can tell from the paper). Perhaps you can add a small comment if you agree with this.

*Reply: Thanks for this suggestion. We are, however, not sure how the particle size distribution (PSD) will exactly evolve after an eruption. In another one of our recent papers (Wrana et al., 2023; reference provided below) we investigated the evolution of the PSD after the eruptions of Ambae (2018), Ulawun and Raikoke (2019) and La Soufriere (2021). In three of theses cases, the median (and effective) radius of the stratospheric aerosols – as retrieved from SAGE III/ISS solar occultation measurements – decreased rather than increased as a consequence of the eruption. In addition, the reduction in size seen in the observations lasted for about half a year after the eruption of Ulawun. The accompanying model simulation also reproduced the initial decrease in size (effective radius), but after August 2019 the modelled sizes increased again, probably due to coagulation. Also because of the discrepancy between observation and model we are uncertain what role coagulation actually plays and how effective and fast it is to change the PSD significantly. For the Raikoke eruption (2019) an increase in particle size is seen in both observations and model simulations. In summary, we do not really know what exactly could have happened after the Krakatoa 1883 eruption and decided not to mention the hypothesis proposed by the reviewer. But it is certainly a possible interpretation or explanation.*

*Reference:*

*Wrana, F., Niemeier, U., Thomason, L. W., Wallis, S., and von Savigny, C.: Stratospheric aerosol size reduction after volcanic eruptions, Atmos. Chem. Phys., 23, 9725–9743, doi.org/10.5194/acp-23-9725-2023, 2023.*

**Comment:** In Fig. 3, the phenomenon of purple light that is often observed after large volcanic eruptions is also simulated, but it is not mentioned in the paper. A small description (one sentence or so) of this finding would enhance the credibility of the method even more.

*Reply: Another good point, thanks for pointing this out. We added the following sentence to the list of conclusions in section 3:*

*"For many scenarios with enhanced stratospheric aerosol loading, purple colours above the green, yellow or orange (depending on the scenario) colours are simulated, as shown in Fig. 4, i.e. the simulations also allow reproducing this well-known phenomenon frequently observed after volcanic eruptions."*

**Technical Corrections**

**Comment:** Line 3: volcanic activity of Krakatau started already several months before August 27. It is perhaps better to speak about the eruption climax of August 27, or something similar.

*Reply: Good point. We decided to remove the exact date from the abstract, because otherwise the sentence would suggest that the green sunsets were observed immediately after the eruption. The sentence now reads:*

*"However, many eyewitnesses reported remarkably clear and distinct green twilight colours after the eruption of Krakatoa (Sunda Strait, Indonesia) in 1883."*

**Comment:** Line 70: This sentence seems to be incorrect. Suggestion: 'Please note that …, while the displayed …'.

    *Reply: Changed.*

**Comment:** Figure 3: the caption seems to be wrong. Panel (b) shows Total Ozone Column, Panel (c) Aerosol Optical Depth, while the caption indicates the reverse.

    *Reply: Thanks for catching this – corrected.*

**Reply to comments by anonymous reviewer #2**

**Comment:** The paper describes a small study which explains why there can be green sunsets after volcanic eruptions. Radiative transfer model simulations are performed for various different aerosol assumptions to explain the green color and to also constrain stratospheric aerosol parameters, i.e. size distribution and aerosol optical thickness. The detailed discussion shows that the found parameters are reasonable. The authors suggest that the method could be used to constrain aerosol properties of historic eruptions, however I suppose that this would be difficult because there are not many reports about the sky colors available and of course also no quantitative observations.

The paper is clearly presented and since it explains the phenomenon of green sunsets to my knowledge for the first time I recommend to publish the paper after very minor corrections (see below).

    *Reply: We thank the reviewer for his/her positive assessment of our manuscript!*

**Comment:** Often, the term "anomalous scattering" is used. I think that this is misleading because the dependece of the scattering coefficient on wavelength is just a result of well-known Mie theory and not "anomalous".

    *Reply: We agree that "anomalous scattering" may be misleading, but this is a standard technical term for the case, where the scattering cross section increases with increasing wavelength. It has been used in the past (e.g. Porch et al., 1989) and was not introduced by us. For this reason we would like to keep the term. However, in order consider the reviewer's comment and to clarify this point, we added the following sentences to the introduction:*

    *"Note that the term "anomalous scattering" only refers to the untypical spectral dependence of the scattering cross section in the visible spectral range. The scattering cross section can, however, still be modelled with, e.g. Mie theory."*

    *Reference:*

    *Porch, W. M., Blue moons and large fires, Appl. Opt., 28, 10, 1989.*

**Comment:** It is not mentioned how the optical properties of the aerosols were computed. I assume Mie theory?

> *Reply: The reviewer is right, this was not mentioned. Instead, we did refer to earlier papers based on a similar approach. But it is certainly an important piece of information and we added the following sentence to section 2:*
>
> *"Note that Mie theory was used to calculate the optical properties of the aerosols."*

**Comment:** The term "volcanic sunset" in the title sounds strange. To me "Explaining the green sunsets after the 1883 volcanic eruption of Krakatoa" sounds better.

> *Reply: OK, changed.*

---

## Author Response (AR3)

**Reply to editor comment by Dr. Timothy Garrett**

Dear Dr. von Savigny,

I am pleased to accept your fascinating paper on green sunsets from stratospheric aerosol associated with the Krakatoa eruption, a wonderful case study in atmospheric optics. My appreciation for drawing a contrast with green thunderclouds sometimes seen at sunset. The point appears to be that the phenomenon addressed here owes to single rather than multiple scattering by large aerosols with a substantial but not too substantial optical depth.

Please consider acknowledging the anonymous reviewers and note the expression in the abstract "To our best knowledge" is better as "To the best of our knowledge"

Best regards,

Tim Garrett

*Reply: Thank you very much for your encouraging comments and for suggesting to include a discussion on green thunderstorms.*

*Following your suggestion we replaced "To our best knowledge" by "To the best of our knowledge" in the abstract.*

*We also added the following sentence to the acknowledgements: "We also thank Drs. Filip Vanhellemont and Timothy Garrett as well as an anonymous reviewer their insightful comments that helped to improve the manuscript."*

*Best wishes, Christian von Savigny*